American Society for Microbiology

# Protection Efficacy of Monoclonal Antibodies Targeting Different Regions of Specific SzM Protein from Swine-Isolated *Streptococcus equi* ssp. *zooepidemicus* Strains

Haoshuai Song,[a,b] Chen Yuan,[a,b] Yu Zhang,[a,b] Fei Pan,[a,b] Hongjie Fan,[a,b,c] Zhe Ma[a,b,c]

[a]MOE Joint International Research Laboratory of Animal Health and Food Safety, College of Veterinary Medicine, Nanjing Agricultural University, Nanjing, China
[b]Ministry of Agriculture Key Laboratory of Animal Bacteriology, Nanjing, China
[c]Jiangsu Co-innovation Center for Prevention and Control of Important Animal Infectious Diseases and Zoonoses, Yangzhou, China

Haoshuai Song and Chen Yuan contributed equally to this work. Author order was determined by last name.

**ABSTRACT** *Streptococcus equi* subsp. *zooepidemicus* (SEZ) has a wide host spectrum, including humans and domestic animals. The SEZ-caused swine streptococcicosis outbreak has occurred in several countries, and the swine-isolated strains usually have specific *S. zooepidemicus* M-like (*szm*) gene types. In this study, we found that the production of this specific *szm* gene (SzM protein) was an effective vaccine candidate. It could provide better protection with a 7-day interval immune procedure than the traditional vaccine strain ST171 and attenuate the strain ΔsezV against swine-isolated hypervirulent SEZ infections. According to this outcome, we developed monoclonal antibodies (McAbs) targeting the variable and conserved regions of this SzM protein, respectively. These McAbs all belong to the IgG1 isotype with a κ type light chain and have opsonophagocytic activity rather than agglutination or complement activation functions. We estimated the protection efficiency of the McAbs with 3 different passive immunotherapy programs. The anti-conserved region McAb can provide effective protection against swine-isolated SEZ infections with only the inconvenient immunotherapy program. It also partially works in preventing infection by other SEZ strains. In contrast, the anti-variable region McAb is only adapted to protect the host against a specific *szm* type SEZ strain isolated from pigs, but it is flexible for different immunotherapy programs. These data provide further information to guide the development of derived, genetically engineered McAbs that have potential applications in protecting hosts against swine-isolated, hypervirulent SEZ infections in the future.

**IMPORTANCE** The swine-isolated SEZ, with its specific *szm* gene sequence, has impacted the pig feeding industry in China and North America and has led to serious economic loss. Though the SzM protein of SEZ has been proven to be an effective vaccine in preventing infection, most previous studies focused on horse-isolated strains, which have different *szm* gene types compared to swine-isolated strains. In this study, we developed the McAbs targeting the conserved and variable regions of this SzM protein from the swine-isolated hypervirulent strains and evaluated their protection efficiency. Our research provided information for the development of chimeric McAbs or other genetically engineered McAbs that have potential applications in protecting pigs against hypervirulent SEZ infections in the future.

**KEYWORDS** *Streptococcus equi* subsp. *zooepidemicus*, SzM protein, swine streptococcicosis, immunotherapy, monoclonal antibody

Address correspondence to Zhe Ma, mazhe@njau.edu.cn.

The authors declare no conflict of interest.

**S**treptococcus equi subsp. *zooepidemicus* (SEZ) belongs to the Lancefield group C streptococci (GCS). It is often isolated from the reproductive and respiratory tracts of horses, and it is usually identified as an opportunistic pathogen (1). SEZ can infect a

wide spectrum of hosts, and it has been acknowledged as a zoonotic pathogen among humans and a variety of animals (2–6). Its infectious symptoms include sepsis, meningitis, endocarditis, and arthritis (7, 8). The most serious impact caused by SEZ infection has happened in the pig feeding industry. In 1976, more than 300,000 pigs died in the Sichuan province of China because of a SEZ pandemic, in which the isolated pathogen was the ATCC 35246 strain. Even now, sporadic outbreaks are still ongoing in China (9, 10). Between 2019 and 2020, high mortality events due to SEZ infections in swine were reported in North America for the first time. The mortality rate could reach 30% to 50% over a period of 2 weeks after infection. In Ohio and Tennessee, strains OH-71905 and TN-74097 were identified as pathogens of this pandemic, respectively. They have an average nucleotide identity of more than 99.9% with ATCC 35246 within the 99.36% aligned sequence length, indicating that these 3 strains are highly homologous. The more important and interesting finding is that these 3 strains share a 100% identity SzM protein sequence, whereas their SzM homology to other strains isolated from horses and other animals is low, especially in the SzM amino terminus (N terminus) variable region (11–13).

SzM protein is a dimeric, coiled-coil, cell membrane-anchored protein on the surface of SEZ with its carboxy terminus (C terminus) LPXTG motif. It is categorized into the M protein family due to its similar structure to the vital virulent factor of the *Streptococcus pyogenes* M protein (14). The N termini of SzM proteins from different strains usually show high heterogeneity (and are named the variable region). In contrast, the C terminus of SzM is always conserved in most SEZ strains (and is named the conserved region) (15). However, the whole length SzM protein sequence homology among different SEZ strains ranges from 19.66% to 54.66% due to the highly variable region, so this protein can be used for the SzM-type categorization of SEZ (13). The ATCC 35246, OH-71905, and TN-74097 strains belong to the same SzM-type, and, according to the serious consequences of their pandemic in the pig feeding industry, we believe that this SzM-type SEZ is a hypervirulent cluster to pigs (16).

The virulence is highly attenuated in *szm* gene defective SEZ because this surface protein is crucial for the survival of SEZ in human and animal blood (15, 17, 18). The variable region also leads to the diverse fibrinogen affinity of the SzM protein, which has been identified as a key character of SzM, to confer an anti-phagocytosis capability to SEZ (18). Previous research has developed the SzM protein as a vaccine candidate for protecting horses from SEZ-caused equine respiratory disease (13). However, the SEZ strains used in those studies were mostly isolated from horses, and these strains share low SzM protein homology with the epidemic strains isolated from infected pigs in China and the U.S. Beyond the horse-isolated strains, the swine-isolated strains have been proven to be hypervirulent to pigs (19). It is necessary to develop a derived SzM protein vaccine against the swine-isolated strains so as to provide protection to pigs. In addition to the vaccine, as a prevention strategy, monoclonal antibody (McAb)-based therapeutics should be more effective at reducing the SEZ-caused sudden death ratio in swine after an infection has occurred (11). According to the homology feature of the SzM protein, the McAb development strategies can be divided into two main categories. Whether the N-terminal variable region or the C-terminal conserved region can provide a more protective antigen epitope is still unknown.

In this study, we verified the protective effect of the recombinant SzM protein in a mouse model. This protein was expressed according to the *szm* gene of the epidemic SEZ strain ATCC 35246, which is 100% identical to the *szm* gene of the epidemic SEZ strains OH-71905 and TN-74097 in the U.S. The McAbs against the conserved and variable regions of the SzM protein were both prepared, and their characterization and protective immunogenicity against SEZ infection were estimated. These data would be helpful in the further development of the genetically engineered McAbs derived from this study, and these McAbs could be applied to protect hosts against hypervirulent SEZ infections in the future.

## RESULTS

**Recombinant SzM protein of the SEZ ATCC 35246 strain provides effective protection against SEZ infection to mice.** To estimate the protective activity of the SzM protein of SEZ ATCC 35246, the truncated SzM protein (tSzM, without a signal peptide and LPXTG motif) (Fig. S1A) was expressed and purified for immunization in a mouse model. The traditional SEZ attenuated vaccine strain ST171 used in China and the artificially constructed ΔsezV strain, which has a defect in the *szm* gene-specific upregulator gene *sezV* and cannot express the SzM protein (15), were used as control immunogens. The mice in each group were immunized three times with the tSzM protein, attenuated live bacteria ST171, and ΔsezV, respectively, at 7-day intervals prior to a challenge (Fig. 1A). Sera were collected from each mouse 5 days after each immunization to detect antibody titers, which are strong clinical indicators of protection against infection. All 3 immunogens can boost the production of antibodies against ATCC 35246 or its SzM protein, and the titers increase after each immunization. Notably, the ST171 immunized mice serum showed lower antibody titers than did the ΔsezV immunized serum after the third immunization, indicating that ST171 may not be an ideal vaccine (at least with this 7-day interval immunization procedure) against a high dosage (>100 times the lethal dosage, $1 \times 10^4$ CFU, which is the minimum dosage that may cause 100% death in mice) hypervirulent SEZ strain (Fig. 1B).

The challenge data from the mice indicated that ST171 and ΔsezV cannot provide effective protection to mice against a high-dosage ($9 \times 10^6$ CFU in this case) SEZ ATCC 35246 infection with the employed 7-day interval immunization procedure. Their protection efficiency was almost as low as that of the PBS mock group. In contrast, even under the high-dosage challenge condition, tSzM-immunized mice still gained 80% protection, which is significantly higher than those of the ST171 (30%, $P = 0.0022$) and ΔsezV (40%, $P = 0.0178$) immunization groups (Fig. 1C). After challenging the tSzM immunized mice with SEZ ATCC 35246, we detected the bacterial burdens in different organs of the surviving mice on days 1 and 5 after inoculation. The tSzM protein prevented the invasion of SEZ ATCC 35246 to most of the organs in the mice. Bacterial infection to the spleen, kidney, and brain was completely impeded. In the lung and blood, only around one-third of the mice still had bacterial colonization in the tSzM-immunized group, while this ratio was almost 100% in the PBS mock group. However, 100% of the mice had bacterial colonization in the liver, both in the PBS and tSzM-immunized groups. According to the surviving percentage, the mice were probably tolerant to the bacterial burden in the liver, which could trap the bacteria for killing by immigrating neutrophils (20) (Fig. 1D). These results suggest that the SzM protein is an effective immunogen in protecting mice from a high-dosage hypervirulent SEZ challenge. This protein could be a promising target for the development of therapeutic antibodies that could be used to prevent or cure hypervirulent, SEZ strain-caused swine streptococcal disease.

**Preparation and characteristic identification of McAbs against the different regions of the SzM protein of SEZ ATCC 35246.** It has been reported that the M protein family members may mainly induce two kinds of antibodies, according to their variable and conserved regions. As the N terminus variable region is heterogeneous in different SEZ strains, antibodies against this region should be specific to ATCC 35246 or other hypervirulent strains that share the same *szm* gene. On the other hand, the antibodies against the conserved region should be promiscuous to most SEZ strains (21). Although the antibodies against the conserved region are promising with respect to conferring wide spectrum protection to the host by hampering infection by most SEZ strains, the variable region is exposed, and it is more feasible for antibodies to attach there than to the C terminus conserved region. Accordingly, we decided to prepare both kinds of antibodies. The N terminus variable region SzM protein (vSzM) and C terminus conserved region SzM protein (cSzM) were expressed and purified as antigens for McAb preparation (Fig. S1B). Their His or GST-tags were identified via Western blot, as well (Fig. S1C). We obtained 3 McAbs for each antigen: anti-cSzM-1 to -3 against cSzM, and anti-vSzM-1 to -3 against vSzM. All 6 of these McAbs were purified

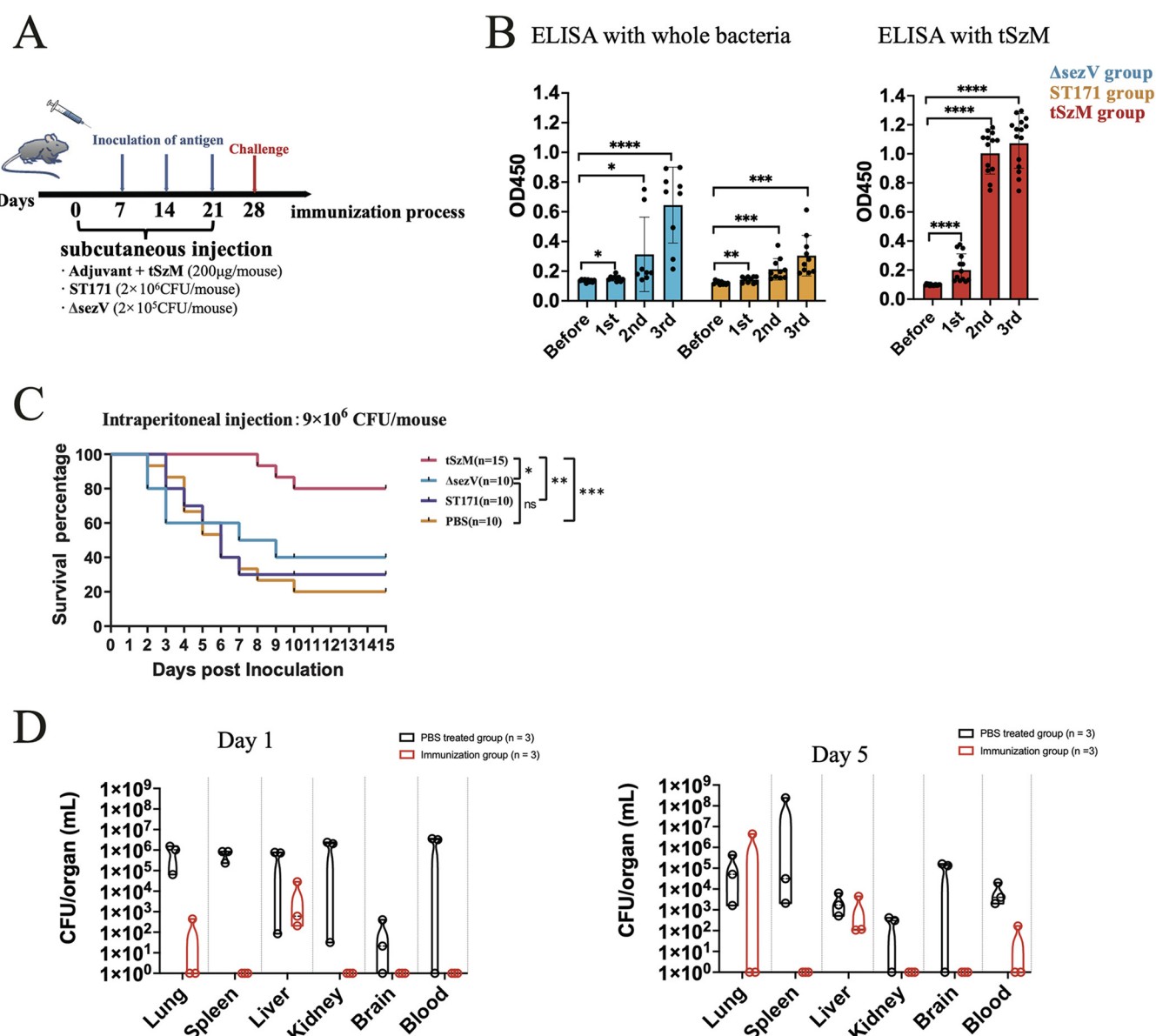

**FIG 1** Recombinant SzM protein protects mice from SEZ ATCC 35246 infection. (A) The diagram of the subcutaneous immunization and challenge process for 3 groups of C57BL/6 mice. The concentration of immunogen is the dosage used for one shot of each immune injection. The blue arrows indicate immunization time points, and the red arrow is the time point of the challenge. (B) Measurement of antibody titer in serum on day 5 after each immunization by indirect ELISA. The ATCC 35246 bacteria and tSzM protein were used as coated antigens in ELISA, respectively. The serum collected from the mice before immunization was used as the negative control. An unpaired Student's $t$ test was used for the statistical analysis of the data. *, $P < 0.05$; **, $P < 0.01$; ***, $P < 0.001$; and ****, $P < 0.0001$. (C) Survival curves of immunized mice after an SEZ challenge. The C57BL/6 mice were challenged with $9 \times 10^6$ CFU ATCC 35246 intraperitoneally on day 7 after the third immunization. A log-rank test was used for the statistical analysis of the data. (*, $P < 0.05$; **, $P < 0.01$; ***, $P < 0.001$; and ns, not significant). (D) The bacterial burden in organs of tSzM protein immunized or placebo group (PBS) C57BL/6 mice on day 1 and day 5 after a challenge with $9 \times 10^6$ CFU ATCC 35246 intraperitoneally.

from mouse ascites and detected with SDS-PAGE for their purity (Fig. S2A) after the assessment of titers (Fig. S2B). The Western blot results showed that all of the McAbs were specific to their target region (Fig. 2A). All 6 of the McAbs belonged to subclass IgG1 isotype with a $\kappa$ type light chain, which is a major light chain type in mice (22) (Fig. 2B). As the SzM protein is primarily located on the surface of SEZ, we can see a green halo around bacteria in the immunofluorescence assay (IFA) after incubating the wild-type (WT) ATCC 35246 strain with each of these McAbs and a fluorescent-labeled secondary antibody, whereas the *szm* gene defective strain was completely in darkness, suggesting that these McAbs were able to bind to natural SzM proteins on intact SEZ surfaces (Fig. 2C; Fig. S3).

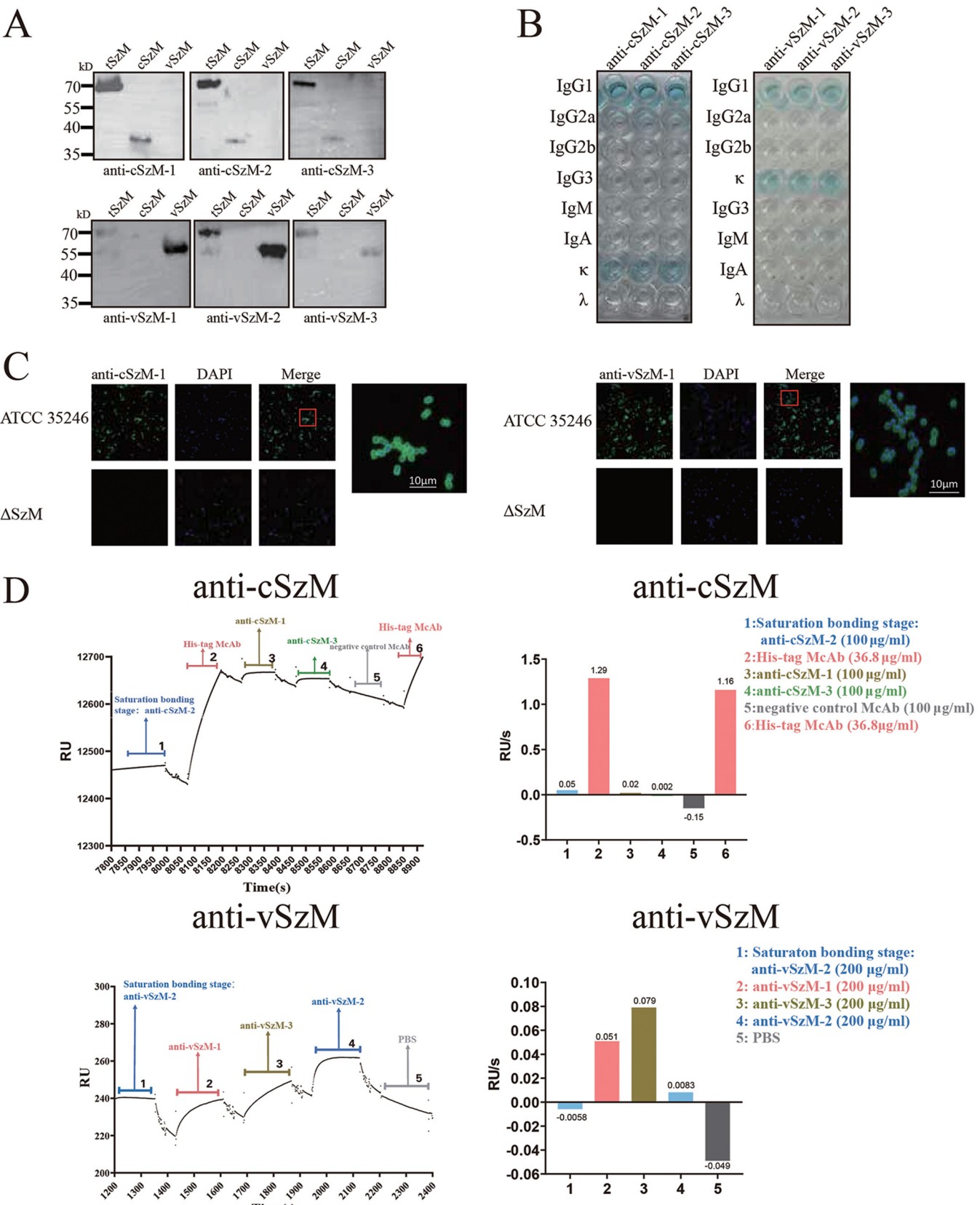

**FIG 2** Characterizations of 6 McAbs against SzM protein. (A) The reaction of 6 McAbs to the tSzM, cSZM, and vSzM recombinant proteins in a Western blot. (B) Identification of the subclasses of the 6 McAbs. The blue wells indicate the corresponding antibody subclasses. (C) Immunofluorescence assay of ATCC

**TABLE 1** Kinetic and affinity characterization of McAbs to the SzM protein[a]

| Name | Ka (1/Ms) | Kd (1/s) | KD (M) |
|---|---|---|---|
| Anti-cSzM-1 | $1.002 \times 10^5$ | $4.753 \times 10^{-4}$ | $4.745 \times 10^{-9}$ |
| Anti-cSzM-2 | $9.789 \times 10^4$ | $4.684 \times 10^{-4}$ | $4.785 \times 10^{-9}$ |
| Anti-cSzM-3 | $1.042 \times 10^5$ | $4.342 \times 10^{-4}$ | $4.166 \times 10^{-9}$ |
| Anti-vSzM-1 | $3.830 \times 10^4$ | $9.092 \times 10^{-4}$ | $2.374 \times 10^{-8}$ |
| Anti-vSzM-2 | $7.652 \times 10^4$ | $1.394 \times 10^{-3}$ | $1.822 \times 10^{-8}$ |
| Anti-vSzM-3 | $1.638 \times 10^4$ | $7.053 \times 10^{-4}$ | $4.305 \times 10^{-8}$ |

[a]Ka, association rate constant; Kd, dissociation rate constant; KD, dissociation equilibrium constant.

To distinguish the epitopes of these 6 McAbs, we employed surface-plasmon resonance (SPR) technology to assess the hindrance of these McAbs to each other during loading in order. If antibodies shared the same epitope, the prior one loaded onto the SPR platform should saturate the epitope and lead to the latter one presenting a constant RU signal, whereas the antibodies against different epitopes can give increasing RU signals along with sample loading (23, 24). After saturating tSzM with anti-cSzM-2, neither of the following anti-cSzM-1 and -3 antibodies increased the RU signal. Only the anti-His tag antibody, which can bind to tSzM through its epitope in the His tag, showed an elevation of the RU signal. However, after saturating the tSzM epitope with anti-vSzM-2, both anti-vSzM-1 and -3 can still lead to an increase in the RU signal. Because the negative-control antibody and PBS had no affinity to tSzM, they only washed the binding antibodies away and presented the minus RU change rate (Fig. 2D). These results indicated that all 3 of the anti-cSzM McAbs recognized the same epitope (or, at least, a similar epitope), whereas the epitopes recognized by the 3 anti-vSzM McAbs were probably distinct, which could also be speculated by the diverse affinity constant (KD) value estimated via SPR. All 3 anti-cSzM McAbs had close affinity constants (KD values), yet the KD values of the anti-vSzM McAbs were more discrete. These data suggest that the anti-cSzM McAbs were more affinitive to tSzM compared to the anti-vSzM McAbs, among which anti-vSzM-2 had the best affinity (Table 1). This could be due to their characteristic diversity of binding epitopes. Overall, the anti-cSzM-1 McAb was chosen as the representative one of the 3 anti-cSzM antibodies, and all 3 of the anti-vSzM McAbs were chosen for use in further investigations.

**McAbs against SzM have opsonophagocytic activity rather than agglutination or complement activation functions.** To assess the bactericidal functions of the above 4 anti-SzM McAbs candidates, we estimated their agglutination activity to SEZ ATCC 35246, complement activation function for lysing bacteria, and opsonization to macrophage phagocytosis. When incubating these 4 McAbs with SEZ ATCC 35246, respectively, none of them showed higher agglutination activity than did the PBS treated group. Although the average $OD_{600}$ value at 3 h was lower in the McAbs-treated ΔSzM group than in the wild-type ATCC 35246, it was likely due to the lack of negatively-charged SzM on the bacterial surface (25), and these mutants tend to agglutinate together over time (Fig. 3A).

To ensure that the complement system in the serum was activatable, we used 2% sheep red blood cells, rabbit anti-sheep erythrocyte antibody, and guinea pig serum as positive-controls and observed red blood cell lysis via antibody-mediated complement activation (Fig. S4). In the antibody-mediated complement bactericidal assay, we found that these 4 anti-SzM McAbs not able to activate the complement system to lyse SEZ and that the polyclonal antibodies (PcAb) from SzM immunized mice or rabbits had no complement activation function (Fig. 3B). Moreover, we checked the C3b deposition on bacterial surface to evaluate the C3 activation capabilities of the anti-cSzM and anti-vSzM McAbs. After incubation with serum, C3b deposition was detected on both SEZ and

**FIG 2** Legend (Continued)

35246 wild-type strain with anti-cSzM-1 and anti-vSzM-1 antibodies. The ΔSzM mutant was used as a negative control. The red rectangle area is zoomed in and displayed on the right side. (D) Identification of the heterogeneity of epitopes in SzM that could be recognized by McAbs by the SPR technology. When the binding between McAbs and the SzM protein happens, the RU/s value will be high. Otherwise, it will be close to zero or negative. Real-time binding RU signal of SPR of 3 McAbs against cSzM (top left). Real-time binding RU signal of SPR of 3 McAbs against vSzM (bottom left). The binding signal value changes ratio in the last two-thirds of the time range during each treatment antibody loading (top right and bottom right).

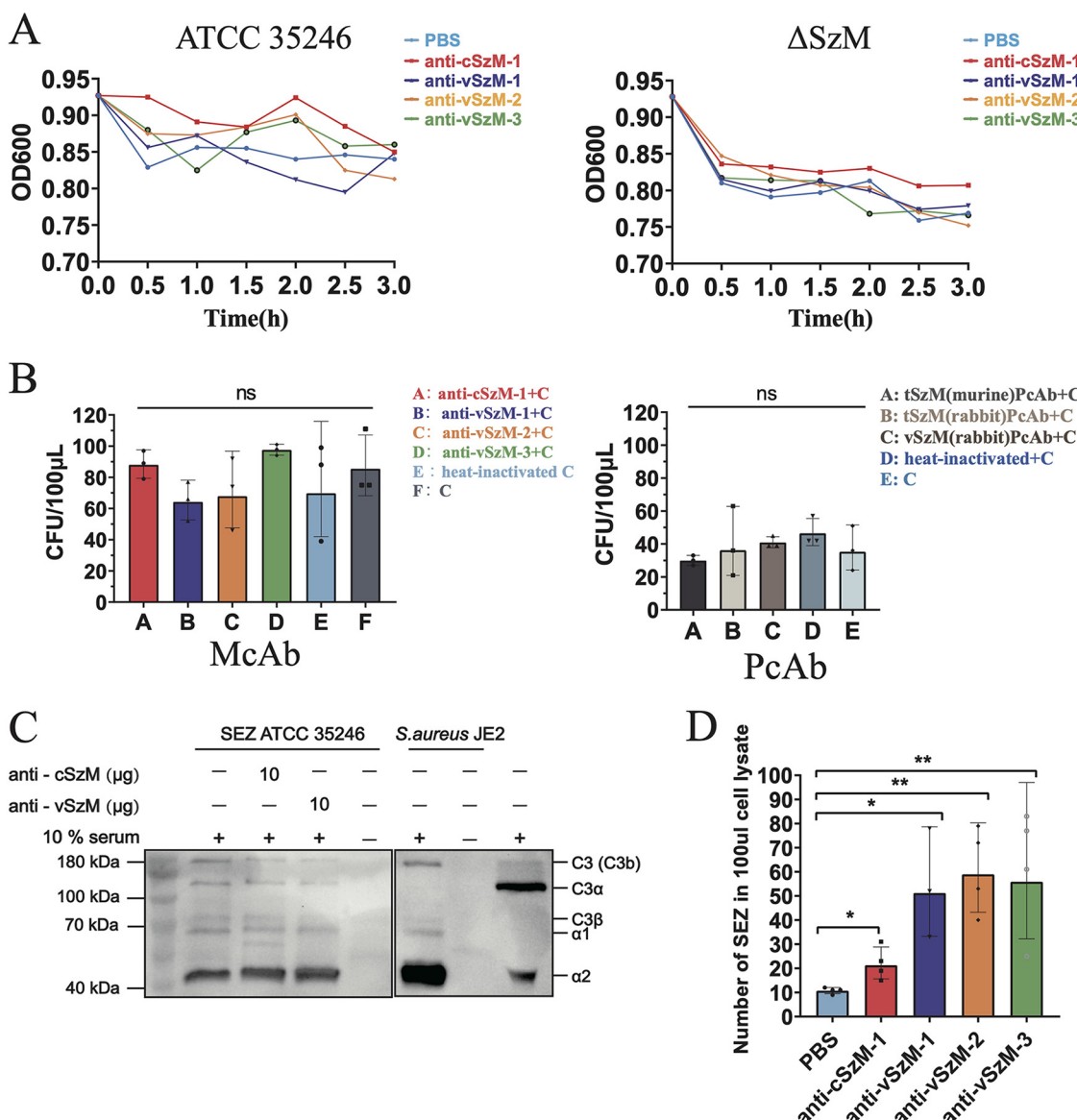

**FIG 3** The biological functions of McAbs. (A) The agglutination test of McAbs against SEZ. The trend of the $OD_{600}$ value represents the agglutination level of SEZ during incubation with McAbs. (B) Antibody-mediated complement bactericidal test. The CFU indicates the survival bacterial number after the treatment of McAbs and the complement. The C in the legend stands for the complement, and the group F is the negative-control which has the complement only. Three replicates were conducted for each McAbs group. (An unpaired Student's $t$ test was used for the statistical analysis of the data. ns, not significant). (C) Western blot analysis of C3b deposition on SEZ. Bacteria were incubated with 10% serum in GVB buffer with or without McAbs. *S. aureus* JE2 with no McAbs was used as a control. The molecules derived from C3 (including deposited C3b) were immunoblotted with the recombinant rabbit anti-C3 antibody. (D) Opsonized phagocytosis assay. The SEZ ATCC 35246 was incubated with mouse primary peritoneal macrophages at an MOI of 1:500 with 200 $\mu$g/mL of McAbs for 1.5 h. After the opsonization, the macrophages were lysed and spread on THB plates for CFU counting. Four replicates were conducted per group. An unpaired Student's $t$ test was used for the statistical analysis of the data. *, $P < 0.05$; **, $P < 0.01$.

*Staphylococcus aureus* (*S. aureus*). However, the addition of McAbs failed to augment the deposition of C3b on SEZ, indicating that these two McAbs did not contribute to the deposition of C3b on SEZ (Fig. 3C). Together, we supposed that the agglutination and the complement activation were not the dominant functions of the McAbs for providing protection after a SzM immunization.

Additionally, we estimated the opsonization functions of the 4 McAbs. Compared to the PBS-treated group, all 4 of the McAbs could enhance the phagocytosis of macrophages to ingest SEZ ATCC 35246 (anti-cSzM-1 [$P = 0.0177$], anti-vSzM-1 [$P = 0.0105$],

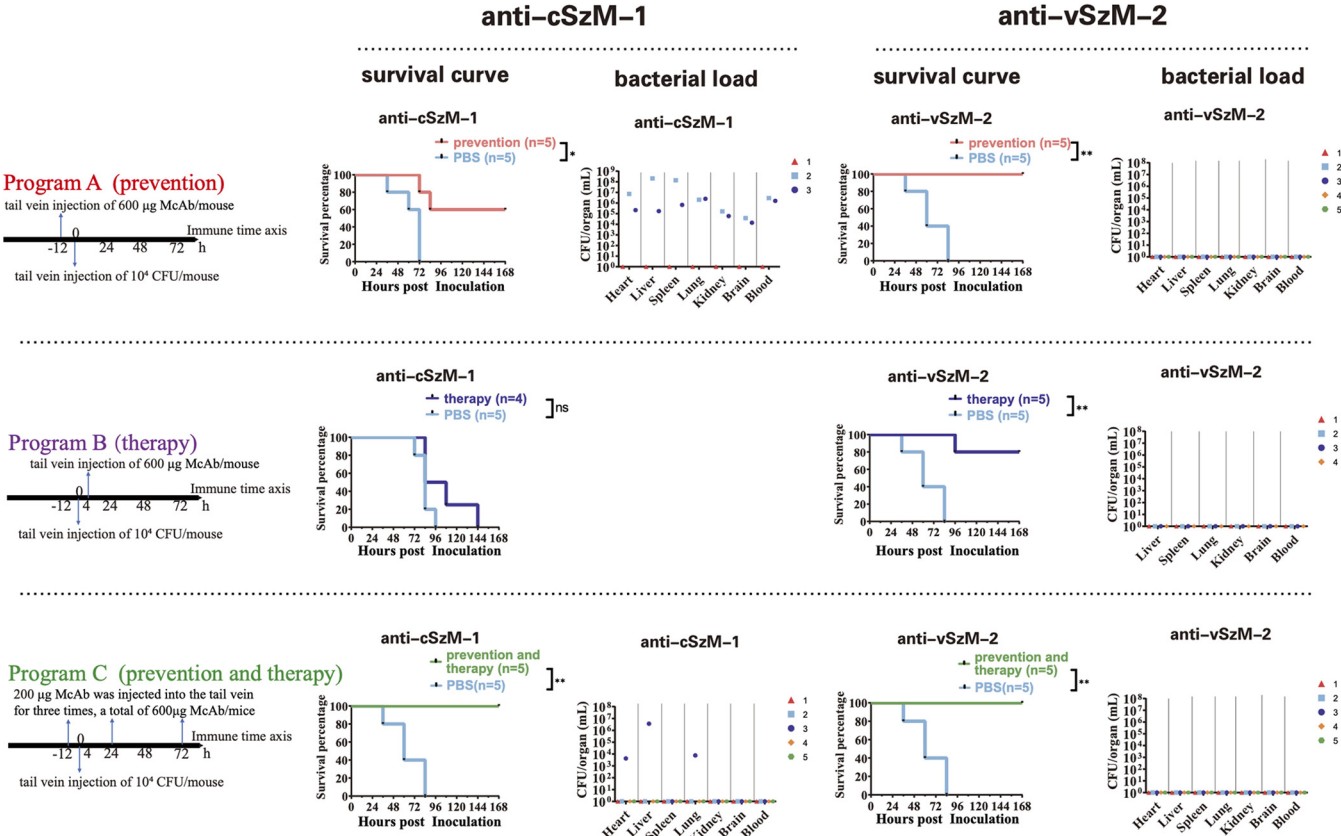

**FIG 4** Protective effect of McAbs against a lethal dosage of ATCC 35246 in mice with 3 different passive immunotherapy programs. The 3 scales show the processes of 3 different immunotherapy programs. The 7-day survival curves of the anti-cSzM-1 and anti-vSzM-2 treated mice with 1 of 3 immunization programs. The corresponding bacterial burdens of the organs of the surviving mice are displayed on the right side. A log-rank test was used for the statistical analysis of the data. (**, $P < 0.01$; ns, not significant).

anti-vSzM-2 [$P = 0.0013$], and anti-vSzM-3 [$P = 0.0081$]). The McAbs against the variable region were even more effective than the one against the conserved region (Fig. 3D). Due to the opsonization estimation results of the anti-vSzM McAbs, we used the most effective one (anti-vSzM-2) and the anti-conserved region McAb (anti-cSzM-1) in a further passive protective immunity assessment.

**The anti-SzM McAbs protect mice from a challenge of a lethal dosage of SEZ ATCC 35246.** With the 3 different passive immunotherapy programs, we estimated the passive protective immunity of the anti-cSzM-1 and anti-vSzM-2 McAbs in a mouse model. Treating the mice with the antibody for prevention at 12 h before the challenge was program A. Treating the mice for therapy with the antibody 4 h after the challenge was program B. Program C included three treatments: a first at 12 h before the challenge, a second at 24 h after the challenge, and a third at 72 h after the challenge. The total amount of McAb used in each program was held constant at 600 $\mu$g per mouse. The anti-cSzM-1 only provided effective protection against a lethal-dosage SEZ challenge when following immunotherapy program C ($P = 0.0026$). All of the mice could survive for at least 7 days, and the bacteria were mostly eliminated from the organs of 80% of the survival mice at day 7 (exception: one mouse still had bacteria in the heart, liver, and lung) (Fig. 4). Surprisingly, the anti-vSzM-2 McAb, which has a lower affinity to the SzM protein than did the anti-cSzM-1, showed excellent protection to mice against a lethal-dosage SEZ challenge in all 3 immunotherapy programs (program A [$P = 0.0026$], program B [$P = 0.0026$], and program C [$P = 0.0026$]). Although one mouse died following program B, the protection rate was still as high as 80% with this program. Moreover, due to the bacterial burdens in organs being undetectable at day 7, we believed that the anti-vSzM-2 McAb was able to facilitate the immunity system to effectively eliminate

SEZ in mice. In conclusion, while anti-cSzM-1 and anti-vSzM-2 McAbs both showed promising protective activity against SEZ hypervirulent strain infections, anti-vSzM-2 was more flexible in passive immunotherapy programs and more effective in SEZ elimination *in vivo*.

**The cross-protection of McAbs against different SzM type strains depends on the homology of the target residue.** To detect the cross-protection of our McAbs, we used them to immunize mice that were challenged by 3 different Lancefield group C streptococci (GCS) strains with diverse M proteins, including two SEZ strains and one *Streptococcus equi* subsp. *equi* (SEE) strain. The virulence of these 3 strains was weaker than the hypervirulent strain SEZ ATCC 35246 to mice (Fig. S5), so we used higher CFU (~100-fold higher than ATCC 35246) to challenge the mice (Fig. 5A, B, and D). The immunotherapy approach followed the most effective (but most complex) program C, which had been proven to be efficient for both the anti-conserved and the variable region antibodies with respect to the protection of the mice. The protective activity of the anti-cSzM-1 antibody against challenges by these 3 GCS strains was much weaker than that observed against SEZ ATCC 35246. Though the protection percentage was only 20% in the SEZ 17006 challenged mouse model, a statistical analysis showed a significant $P$ value ($P = 0.0072$) compared to that of the mock group (Fig. 5A). This McAb completely failed to protect mice from a challenge of SEZ 18055 ($1.3 \times 10^6$ CFU) or SEE 17009 ($1.6 \times 10^6$ CFU) strains (Fig. 5B and D). To reduce the bacterial virulence severity in the mouse model, we decreased the challenge doses of these two strains ~10-fold lower. The anti-cSzM-1 antibody showed significant protective efficiency against SEZ 18055 infection ($P = 0.0116$) at the $9.2 \times 10^4$ CFU challenge dose (Fig. 5C), but it still failed to protect mice from a SEE 17009 challenge with a $8.4 \times 10^4$ CFU dose, as the SEE has a high identity in the conserved region of its SeM that aligns to the SzM protein (15) (Fig. 5E). However, the bacterial burden had not been eliminated in any of the surviving mice, despite the strain types and their dosages used for the challenge (Fig. 5). On the other hand, the anti-vSzM-2 McAb cannot provide significant protection to any of these 3 GCS strain-infected mouse models, with most of them still ultimately tending to 100% death, suggesting that anti-vSzM-2 only protects against its specific targeting SzM type ATCC 35246 strain due to the diversity of the variable region. Overall, these data suggest that the protective spectrum of the anti-SzM antibody could partially depends on the homology of its target region. Although the anti-conserved region antibody can provide protection against different SEZ strains, its complex immunotherapy program was less accessible. So, developing anti-variable region antibodies according to a specific SzM type is a more effective means for the prevention and therapy of SEZ, including GCS infection.

## DISCUSSION

Monoclonal antibody (McAb) therapy has been developed rapidly in the bacterial infectious diseases field, especially as traditional antibiotics are becoming increasingly obsolete due to antimicrobial resistance. Several McAbs have been approved by the Food and Drug Administration (FDA), and the number of promising clinical trials is growing and includes trials of McAb against Gram-positive bacterial surface proteins, such as protein A of *Staphylococcus aureus* and the M protein of *Streptococcus pyogenes* (26, 27). McAb against a surface target usually depends on the strategy of disabling bacterial immune evasion. The SzM protein, used as a McAb target, is known to participate in fibrinogen binding and crucial for SEZ to survive in the blood (18). Moreover, the SzM protein from horse isolated strains have been proven to be able to provide effective protection, consistent with the SzM protein of this study, suggesting that it could be an ideal target for McAb development for the prevention and treatment of SEZ infections (14). There have been several studies that prepared McAbs against the M family proteins of SEE and estimated their protection in mouse models (28, 29). However, M family proteins of SEE have a distinct N terminus variable region to SzM. SEE has strict host restrictions to the horse, and the McAb against this region might not hinder a hypervirulent SEZ strain (with its specific SzM type) to infect a host. Thus,

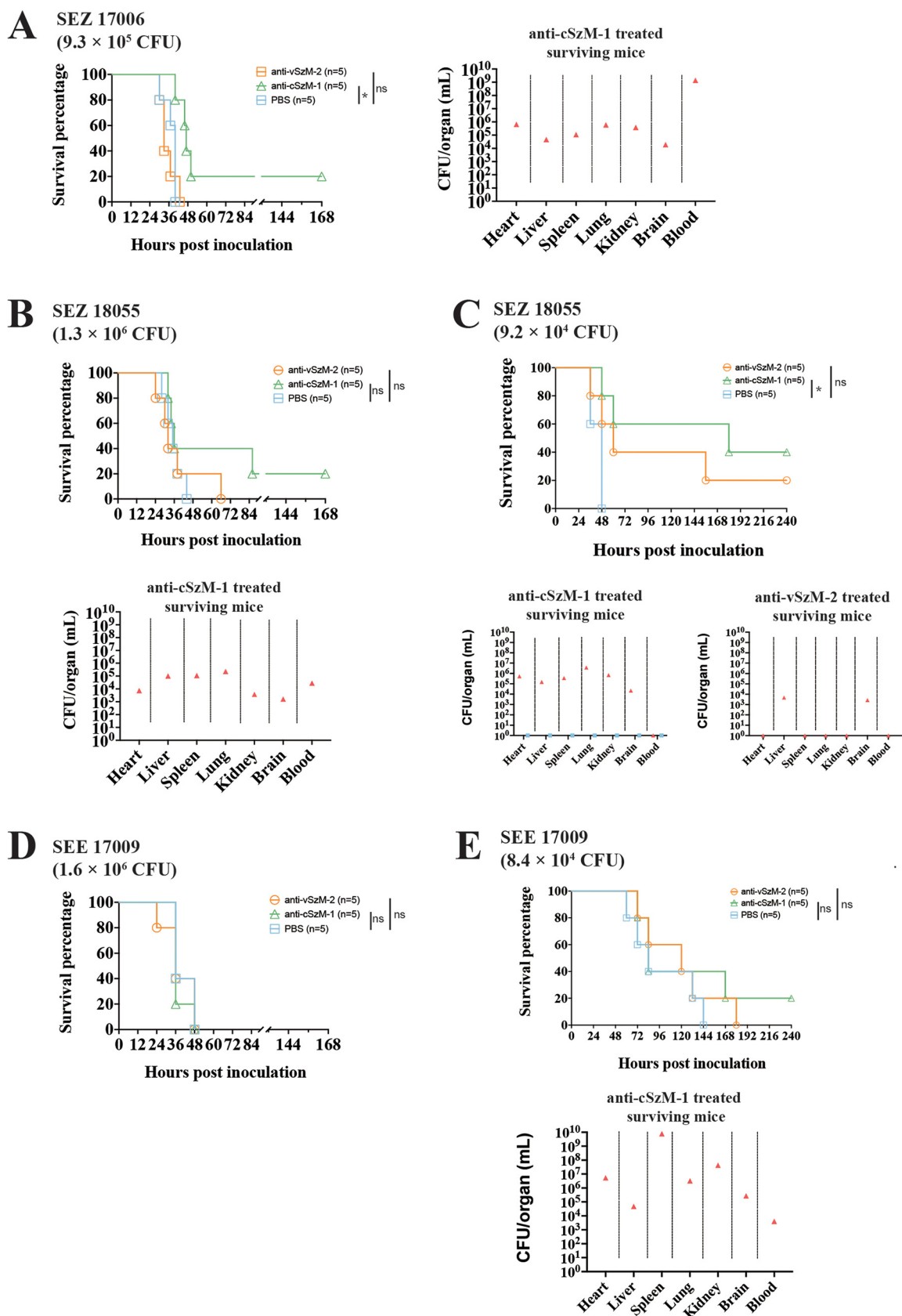

**FIG 5** The cross-protection of McAb against different GCS strains in C57BL/6 mice. The survival curves of anti-cSzM-1 and anti-vSzM-2 treated mice with immunization program C and corresponding bacterial burdens of the organs of the surviving mice. (A) SEZ

in this study, we targeted not only the conserved region but also the variable region of SzM from swine-isolated hypervirulent SEZ strains for the preparation of McAbs.

Although the traditional vaccine strain ST171 had been used for the prevention of hypervirulent SEZ strain infection in pigs in China in the 1970s and had an identical *szm* gene to that of ATCC 35246, this strain was not protective against the high-dosage SEZ challenge to mice. As this live attenuated vaccine was produced by continuous passaging under thermal stress 171 times, it had a growth defect and may not have reproduced well in the mice, leading to low antibody titers and poor protection after immunization (30). The ΔsezV strain also failed to protect against SEZ infection. It was *sezV* gene defective and had no SzM protein expression but displayed normal growth capability. This strain had low virulence and was worth considering as a potential vaccine (15). However, without the vital protective immunogen SzM protein, even if ΔsezV induced a higher antibody titer than did ST171 immunization in mice, these antibodies, boosted by the rest of the antigens in ΔsezV, could not remedy the shortage.

Currently, therapies based on McAbs are usually costly, so they have not been widely used in treating the infectious diseases of animals. However, for some valuable individuals, such as stud boars or pregnant sows, treatment via McAbs will be worthwhile, especially when antibiotic treatments do not perform well. In addition, along with the technical development of McAbs production, the cost will decrease to an affordable level for the pig feeding industry. Moreover, as a zoonotic pathogen, SEZ caused human death in Thailand at a fatality rate of 42.8% (31), indicating that the McAbs against SEZ may have a potential application in humans after appropriate genetic modification.

The isotype of the McAbs developed in this research are uniform; they all belong to the IgG1 isotype with a κ type light chain. The IgG1 isotype is normally the most abundant subclass and has a high FcγR receptor binding affinity, which leads myeloid cells to identify opsonized bacteria (32, 33). Though we had found that the opsonization activities of these McAbs promote efficient phagocytosis of SEZ, the complement activation and agglutination functions were both absent. It has been suggested that the IgG1 isotype can efficiently trigger the classical route of the complement by binding C1q through its CH2 region (34). However, the role of C1q in the fight against a streptococcal infection is controversial (35). The membrane attack complex (MAC) can mainly kill only Gram-negative bacteria after complement activation (36). Gram-positive bacteria, such as streptococci, which are surrounded by a capsule, can inhibit the activation of the complement (37, 38). At the least, the IgG1 McAbs and PcAbs prepared in this study are not sufficient to activate the complement to lyse hypervirulent SEZ strains coated with a thick capsule. The hyaluronic acid capsule of SEZ could also be the reason that the McAbs failed to aggregate bacteria, as the negative charges tend to repel cells, which has been reported in *S. pyogenes* (39). Besides, the McAb that bound to bacterial surface polysaccharides tended to strongly agglutinate bacteria, in contrast to those that bound to surface proteins (40). Opsonization triggers phagocytosis and intracellular killing by phagocytic cells, and it is effective in killing both Gram-negative and Gram-positive bacteria. The McAbs in this study eliminated SEZ predominantly via their antibody-mediated opsonophagocytosis activities.

Usually, the McAbs were administered to the host for rapid protection following a known exposure to pathogens (41). Pandey et al. evaluated their McAbs against GAS (Lancefield group A streptococci) M protein and exotoxin Spe by administering McAbs 18 h after a streptococcal infection (26). However, to prevent bacterial infection in advance, McAbs could be administered before the exposure to the pathogen, as well, such as was observed with the anti-alpha toxin McAb used in the prevention of *S. aureus* induced pneumonia, in which the McAb was administered 24 h before challenge

**FIG 5** Legend (Continued)

17006 with challenge dosage at $9.3 \times 10^5$ CFU. (B) SEZ 18055 with challenge dosage at $1.3 \times 10^6$ CFU. (C) SEZ 18055 with challenge dosage at $9.2 \times 10^4$ CFU. (D) SEE 17009 with challenge dosage at $1.6 \times 10^6$ CFU. (E) SEZ 17006 with challenge dosage at $8.4 \times 10^4$ CFU. A log-rank test was used for the statistical analysis of the data. (*, $P < 0.05$; ns, not significant).

and successfully limited the severity of pneumonia in mice (42). The anti-staphylococcal surface carbohydrate, poly-*N*-acetylglucosamine (PNAG), McAb also could protect against *S. aureus* prior to a bacterial challenge (43). In our passive protective activity assessment of McAbs against SEZ, we conducted 3 passive immunotherapy programs. Interestingly, not all of the McAbs worked well in every program. We have not known the underlying mechanism that anti-cSzM-1 McAb has to be administered both before and post SEZ infection in order to make it effective. Moreover, these programs were evaluated based on mice. So, to administrate these McAbs or their derived genetically engineered antibodies to other hosts, these programs may need modification and reevaluation. The direction for further study may focus on its affinity to Fc$\gamma$R and its relevant opsonic activity with phagocytic cells. The protective efficiency of anti-cSzM-1 McAb is not as effective against other GCS strains as it is against ATCC 35246, and it relies on a complex immunotherapy program. Although it could be considered a potential universal therapeutic antibody against GCS, we still suggest that the most effective means for the prevention of SEZ infection is to develop McAbs according to the variable region of each SzM type strain.

In conclusion, we show that hypervirulent SEZ strain SzM protein vaccination results in a more effective immunity protection than does the traditional vaccine ST171 and the gene knockout attenuate strain ΔsezV in mouse models with a 7-day interval immune procedure. An infection of the brain is closely related to death after a SEZ infection, and the SzM immunization is capable of preventing SEZ colonization in the brain. We prepared 6 IgG1 isotype McAbs against either the variable region or the conserved region of SzM. The McAbs to the conserved region recognize similar epitopes, whereas the epitopes of McAbs to the variable region are diverse. Though none of these McAbs present agglutination or complement activation-dependent bacteria lysis activity, the opsonophagocytosis activity of these McAbs is sufficient for immunotherapy against a SEZ infection to a host. The anti-cSzM-1 has a better protective efficiency against SEZ ATCC 35246 than do other SEZ strains, but it failed to impede SEE infection, which may limit its universal applicability against GCS infection. Moreover, its immunotherapy program is unfriendly for field usage. In contrast, the anti-vSzM-2 with a flexible immunotherapy program results in a satisfactory protection rate as well as the clearance of SEZ from the entire body. The hybridoma cell line that produces anti-vSzM-2 could be used as a potential material in the further development of genetically engineered McAbs to protect hosts from infections of hypervirulent SEZ strains. However, for other SzM type strains, specific McAbs against the variable regions should be developed accordingly.

## MATERIALS AND METHODS

**Strains and culture.** ATCC 35246 was purchased from the American National Standard Preservation Center (ATCC). The SEZ ST171, SEZ 17006, SEZ 18005, and SEE 17009 strains were gifted by Cohen Noah (Texas A&M University). The SEZ ATCC 35246 ΔSzM and SEZ ATCC 35246 ΔsezV strains were constructed by our laboratory. These 7 strains were recovered on Todd Hewitt Broth (THB) plates and grown aerobically to the logarithmic stage ($OD_{600}$ = 0.6 to 0.8) in THB liquid media.

**Mice.** 6 to 8-week-old wild-type (WT) female BALB/c, C57BL/6, ICR mice were housed in the Animal Experiment Center of Nanjing Agricultural University. All of the experimental protocols were approved by the Animal Ethics committee of Nanjing Agricultural University (protocol number: NJAU.No20211112168) and strictly followed the animal experimentation guidelines (GB/T 35892-2018). The immunized or challenged mice were monitored daily. Blood samples were collected via submandibular puncture for immunological assays. The mice were euthanized via isoflurane inhalation followed by cervical dislocation at the end of the study or when they showed clinical signs of moribund condition (impaired mobility).

**Generation of McAbs.** McAbs to the SzM protein were generated via the immunization of BALB/c mice with 200 $\mu$g SzM in complete Freund's adjuvant followed by boosters of 100 $\mu$g of SzM in incomplete Freund's adjuvant. Fusion and cloning were performed as described previously (44). Hybridoma cells secreting specific antibodies were screened via indirect enzyme-linked immunosorbent assay (ELISA).

**The detection of serum-specific antibody levels.** C57BL/6 mice were anesthetized with isoflurane, the serum was collected via the tail vein, and specific antibody levels were assayed via ELISA. SzM protein and ATCC 35246 were diluted in carbonate-bicarbonate buffer and used as coating antigens to detect the serum antibody levels of mice in the SzM immunization group and the attenuated live bacteria immunization groups (ST171, ΔSezV), respectively. For the antigen coating, 100 $\mu$L of ATCC 35246 ($10^7$ CFU/mL) was added to each well, and then an ELISA plate (Corning) was placed in a 56°C oven to

dry the water, while the SzM protein (8 $\mu$g/mL) was coated overnight at 4°C. After coating, the ELISA plate was washed with PBST (PBS + 0.5% Tween 20) 3 times, and then 5% skim milk was added to block the plate at 37°C for 2 h. After blocking, 1:200 diluted serum was added and incubated at 37°C for 1 h, and then 1:2,000 diluted goat anti-mouse IgG (HRP) (Abcam) was added and incubated at 37°C for 45 min. After incubation, 3,3′,5,5′-tetramethylbenzidine (Beyotime, China) was used as a chromogenic substrate. Finally, 100 $\mu$L of 2M $H_2SO_4$ was added to each well to stop the reaction, and the optical density at 450 nm ($OD_{450}$) was read on a microplate reader (Tecan Infinite M200 Pro).

**Challenge experiment of immunized mice.** Each C57BL/6 mouse was injected intraperitoneally with $9 \times 10^6$ CFU of log-phase ATCC 35246 in the estimation of the immune protection of the SzM protein. The survival of the mice was observed over 15 days. To determine the tissue burdens of mice immunized with the SzM protein and PBS on days 1 and 5 after the challenge, the mice were euthanized via isoflurane inhalation and subsequent cervical dislocation, the tissues were homogenized, and serial dilutions of the homogenates were plated on THB plates to enumerate the bacterial CFU. For the bacterial load in the blood, 30 $\mu$L of blood was collected via the tail vein before euthanizing the mice, and serial dilutions were plated on THB media.

**Western blot.** SDS samples were boiled at 95°C for 10 min prior to running on 10% SDS-PAGE gels, and the prestained protein standard (Vazyme) was used as a molecular weight marker. After the transfer, the membranes were blocked with 5% skim milk at 4°C for 12 h. Then, the ascites antibodies (1:500 diluted with PBS) were incubated with the membranes at 37°C for 1.5 h. Horseradish peroxidase (HRP)-conjugated goat anti-mouse IgG antibody (1:5,000; CMCTAG) was incubated with the membranes at 37°C for 1 h. SuperSignal chemiluminescent substrates (Thermo Fisher Scientific) were used as the HRP substrates.

**Identification of the class, subclass, and type of McAbs.** The identification of the class, subclass, and type of the monoclonal antibody was carried out according to the kit instructions (Biodragon, China). The antigen protein was diluted to 1 $\mu$g/mL in carbonate-bicarbonate buffer and then added to the ELISA plate at 4°C for 12 h. Hybridoma cell supernatant was incubated as the primary antibody at 37°C for 30 min. The HRP-conjugated secondary antibody in the kit was applied at 37°C for 30 min, and then 3,3′,5,5′-tetramethylbenzidine (Beyotime, China) was used as a chromogenic substrate.

**Immunofluorescence assay.** The ATCC 35246 and ΔSzM strains were cultured on THB plates at 37°C for 16 h. A small number of colonies were smeared on sterile glass slides and allowed to dry naturally. Then, they were fixed with 4% paraformaldehyde (MDBio) for 10 min; 0.1% Triton X-100 for 10 min; 5% skim milk at 37°C for 1 h; 1:500 diluted ascites for 1 h at 37°C; goat anti-mouse AF488 (1:500, Abcam) for 1 h at 37°C; and 1:10 diluted 4′,6′-diamidino-2-phenylindole, (DAPI, Invitrogen) for 10 min. They were washed with PBST 3 times between the above steps. Finally, a laser confocal microscope (Zeiss LSM780) was used for observation.

**Agglutination test.** ATCC 35246 and ΔSzM were cultured in THB liquid medium, where they grew to log phase ($OD_{600}$ = 0.6 to 0.8), were centrifuged at 7,000 rpm, and were washed 3 times with PBS. Then, the bacterial solution was adjusted to $OD_{600}$ = 1.0. McAb was added to the SEZ ATCC 35246 and ΔSzM bacterial liquid. The final antibody concentration was 200 $\mu$g/mL after fully vortexing and mixing. Then, the bacteria were placed at 4°C. It was crucial for this series of experiments that the bacteria were not shaken or vortexed during incubation. The $OD_{600}$ of the uppermost liquid was measured every 0.5 h for a total of 3 h.

**Complement hemolytic assay.** The complement hemolytic protocol has been described (45). To evaluate the activity of the complement used in this assay, we first added 1.2 $\mu$L of the rabbit anti-sheep erythrocyte antibody (Xinfan, China) to 1 mL of a 2% sheep red blood cell suspension. The mixture was gently mixed and bathed in water at 37°C for 30 min to ensure that the anti-sheep erythrocyte antibody was fully bound to the erythrocyte. As source of complement, 20 $\mu$L of guinea pig serum were added to the mixture, which was then incubated in a water bath at 37°C for 30 min. Hemolysis indicated that the complement had cell lysis activity.

**Antibody-mediated complement sterilization test.** The protocol involved in the experiment has been described (46). When ATCC 35246 in THB liquid medium grew to the logarithmic stage, the bacterial solution was centrifuged at 7,000 rpm for 10 min, and the supernatant was discarded. The bacterial precipitation was washed with sterilized PBS 3 times, and then the concentration of the bacterial solution was adjusted to approximately $10^3$ CFU/mL. 500 $\mu$L of the bacterial solution was put into a 1 mL EP tube, and then the monoclonal antibody or the polyclonal antibody was added into the bacterial solution. The antibody concentration was 200 $\mu$g/mL. After vortex mixing, the bacterial solution was incubated at 37°C for 30 min, and then 125 $\mu$L of guinea pig serum were added to each tube of the mixture at 37°C for 1 h. During incubation, the samples were turned upside down every 15 min. Finally, 100 $\mu$L were taken from each tube of the sample for counting on the THB plate.

**Complement deposition assays.** The SEZ were grown in Todd-Hewitt broth until the $OD_{600\ nm}$ was between 0.6 and 0.8. The bacteria were washed twice with PBS, and the $OD_{600\ nm}$ value was modulated to 0.5 (~$3 \times 10^8$ cells). The adjusted bacteria were divided to 500 $\mu$L per tube and incubated with an antibody at room temperature for 1 h and then centrifuged and resuspended in 500 $\mu$L of 10% serum diluted in gelatin veronal buffer (GVB) with calcium (0.15 mM) and magnesium (0.5 mM). Bacteria incubated with only GVB served as a negative-control. The bacteria were then opsonized at 37°C for 30 min, using the addition of 24 $\mu$L ice-cold PBS buffer containing 0.5 M EDTA to stop the reaction. The bacteria were washed twice in PBS-1% SDS, resuspended in 50 $\mu$L of SDS sample buffer, and boiled for 10 min. Then, 25 $\mu$L of sample were loaded onto SDS-PAGE for electrophoresis. Recombinant rabbit anti-C3 antibody (1: 2,000) was used in subsequent immunoblots to equine C3 and was developed using HRP labeled goat anti rabbit IgG. In these cases, 10% serum and *S. aureus* JE2 were used as controls.

**The obtainment of mouse primary peritoneal macrophages.** The collection of peritoneal macrophages has been described (47). ICR mice were intraperitoneally injected with 1 mL sterilized broth culture medium (1% tryptone, 0.3% beef extract, 0.5% sodium chloride) for 3 consecutive days. Then, the mice were euthanized via isoflurane inhalation and subsequent cervical dislocation and immersed in 75% alcohol for 3 to 5 min. The abdominal skin of mouse was cut, and the peritoneum was fully exposed on an ultraclean table. Then, 5 mL of PBS was injected into the abdominal cavity of the mouse with a 5 mL syringe, and the abdomen of the mouse was gently massaged with alcohol cotton balls for 3 to 5 min. The fluid in the abdominal cavity was then drained and again rinsed 2 to 3 times with PBS. The obtained rinse solution was centrifuged at 1,000 rpm for 10 min. After centrifugation, the supernatant was discarded, and the cells at the bottom of the tube were suspended with DMEM (Gibco). The cell suspension was later added to a 24-well cell culture plate at a volume of 500 $\mu$L per well. After 2 h of incubation in the cell incubator at 37°C and 5% $CO_2$, the cell plates were washed with DMEM 3 times, and then 500 $\mu$L of DMEM containing 10% fetal bovine serum was added to each well. The cell plates were then placed in a cell incubator for 48 h for subsequent experimentation.

**Opsonophagocytosis test.** Opsonophagocytosis experimental protocol has been described (48) and modified on this basis. The 24-well cell plate (Corning) containing the primary mouse peritoneal macrophages was prepared in advance and washed with DMEM (Gibco). Then, 400 $\mu$L of fresh DMEM culture medium and 12.5 $\mu$L of (50 $\mu$g) McAb were added to each well. The cell plate was then incubated in a cell incubator for 1 h. At the same time, the ATCC 35246 bacterial solution in the logarithmic phase was washed with PBS 3 times, and the concentration of the bacterial solution was adjusted to approximately $10^7$ CFU/mL. McAb was added to the bacterial solution and incubated in a ratio of about 100 $\mu$L of bacterial solution to 12.5 $\mu$L of antibody (50 $\mu$g), and the mixture was incubated at 37°C for 30 min. After incubation, the 112.5 $\mu$L mixture solution was added to the cell well (at a multiplicity of infection [MOI] of 500). At this time, the McAb concentration in each well was approximately 200 $\mu$g/mL. After the addition of the bacteria solution, the 24-well cell plate was centrifuged at 1,000 rpm for 10 min to ensure that the bacteria could fully contact the cells, and then the 24-well plate was placed in a cell incubator for 1.5 h. Next, the 24-well plate was washed with PBS 3 times, and a DMEM culture medium containing penicillin (5 $\mu$g/mL) and gentamicin (100 $\mu$g/mL) was added for 1 h. After the incubation of the antibiotics, the 24-well plate was washed 3 times with PBS. 500 $\mu$L of sterilized ultrapure water was added to each well for 10 min to lyse the cells, and then 500 $\mu$L of 2×PBS was added to adjust the osmotic pressure. Finally, a CFU count was performed after the continuous dilution of cell lysates.

**Affinity detection by SPR.** SPR measurements were made using a Biacore T200 SPR system (GE Healthcare). The tSzM protein was immobilized via amine coupling on a CM5 chip (GE Healthcare) for 70 resonance units, according to the manufacturer's instructions. The flow rate of McAb was 30 $\mu$L/min, the binding time of the antibody was 300 s, and the dissociation time was 600 s. The affinity detection was carried out via the multicycle kinetic method. The sensor chip was regenerated with 10 mM glycine (hydrochloric [pH = 3.0]) for each cycle. The final results were fitted using the BIAevaluation software package (1:1 binding). The negative-control McAb was an anti *Lawsonia intracellularis* Omp2 protein antibody prepared by our lab (49). The rate of the RU value change was calculated according to the data acquired in the last two-thirds of the time range of each McAbs treatment (80 s out of 120 s for anti-cSzM and 120 s out of 180 s for anti-vSzM), which could represent the stable stage of the interaction between the two molecules.

**Identification of monoclonal antibody binding epitopes.** The protocol involved in the experiment has been described (50). To identify the McAbs against cSzM, the corresponding epitope of the tSzM protein on the CM5 chip was saturated with anti-CSzM-2 (200 $\mu$g/mL). When the binding signal tended to be stable, samples of a positive-control (36.8 $\mu$g/mL, anti-His McAb), anti-cSzM-1 (100 $\mu$g/mL), anti-cSzM-3 (100 $\mu$g/mL), and a negative-control McAb (100 $\mu$g/mL, McAb against heat shock protein from *Lawsonia intracellularis*) were loaded. Changes to the binding signal were detected for each sample within 1 min prior to the end of the loading. If a new binding signal was present, it indicated that the binding epitope of the antibody was different from that of the anti-cSzM-2. The epitope identification of the McAbs against vSzM was similar to that observed against cSzM McAbs.

**Protective testing of McAbs in mice.** The McAb immunization was divided into three programs. Program A was given 600 $\mu$g of McAb per mouse at 12 h before the challenge. Program B was given 600 $\mu$g McAb per mouse at 4 h after the challenge. Program C was given 200 $\mu$g McAb at 12 h before the challenge as well as 24 h and 72 h after the challenge (such that the total McAb was 600 $\mu$g per mouse). Both the McAb and the bacteria were given by means of an intravenous injection via the tail vein. The challenged doses of ATCC 35246, SEZ 17006, SEZ 18055, and SEE 17009 were $2.4 \times 10^4$ CFU/mouse, $9.3 \times 10^5$ CFU/mouse, $1.3 \times 10^6$ CFU/mouse, and $1.6 \times 10^6$ CFU/mouse, respectively. After a 7-day observation period, the surviving mice were euthanized via isoflurane inhalation and subsequent cervical dislocation, the tissues were homogenized, and serial dilutions of the homogenates were plated on THB media to enumerate the bacterial CFU.

**Statistical analysis.** The statistical tests were performed using GraphPad Prism 8.0.1. Unpaired Student's *t* tests were used for the statistical analysis of the data (such as the serum-specific antibody levels assay, the antibody-mediated complement bactericidal assay, and the opsonized Phagocytosis assay). A log-rank test was used for the animal experiments. A *P* value of less than 0.05 was considered to indicate statistical significance. *, $P < 0.05$; **, $P < 0.01$; ***, $P < 0.001$; ns, not significant.

## SUPPLEMENTAL MATERIAL

Supplemental material is available online only.
**SUPPLEMENTAL FILE 1**, PDF file, 0.5 MB.

## ACKNOWLEDGMENTS

This research was supported by the National Key Research and Development Program of China (2021YFD1800800), the National Natural Science Foundation of China (31973004), the National Key Research and Development Program of China (2021YFD1800404), the Qinglan Project of Jiangsu Province, the Priority Academic Program Development of Jiangsu Higher Education Institutions (PAPD), and the "Young Scholars" cultivation program of the College of Veterinary Medicine in Nanjing Agricultural University.

We sincerely appreciate Cohen Noah for providing the SEE and SEZ strains used in this study.

H. Song, C. Yuan, and Y. Zhang developed the methodology, and performed the formal analysis, investigation, visualization, and data curation. F. Pan assisted with the investigation and visualization. H. Fan and Z. Ma reviewed and edited the final manuscript, acquired funding, and performed supervisory and administrative tasks during the project.

We declare that we have no conflicts of interest.

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
