## [Reviewer comments · Microbiology Spectrum]

Microbiology Spectrum

Protection efficacy of monoclonal antibodies targeting different regions of specific SzM protein from swine-isolated *Streptococcus equi* ssp. *zooepidemicus* strains

Haoshuai Song, Chen Yuan, Yu Zhang, Fei Pan, Hongjie Fan, and Zhe Ma

Corresponding Author(s): Zhe Ma, Nanjing Agricultural University

Review Timeline:

Submission Date:	May 11, 2022
Editorial Decision:	July 11, 2022
Revision Received:	August 23, 2022
Accepted:	September 27, 2022

Editor: Artem Rogovskyy

Reviewer(s): The reviewers have opted to remain anonymous.

Transaction Report:

DOI: <https://doi.org/10.1128/spectrum.01742-22>

July 11, 2022

Dr. Zhe Ma
Nanjing Agricultural University
Nanjing
China

Re: Spectrum01742-22 (Protection efficacy of monoclonal antibodies targeting different regions of specific SzM protein from swine-isolated *Streptococcus equi* ssp. *zooepidemicus* strains)

Dear Dr. Zhe Ma:

Link Not Available

Sincerely,

Artem Rogovskyy

Journals Department
Reviewer comments:

Reviewer #1 (Comments for the Author):

This manuscript describes the use of SzM and anti-SzM monoclonal antibodies against the variable and conserved domains of this protein to protect mice from infection with a *S. zooepidemicus* strain that was recovered from a case of infection in pigs. Variations of this protein have been shown previously to protect mice, but not horses from infection with *Streptococcus zooepidemicus* or *Streptococcus equi*. The application of monoclonal antibodies to treat infected animals is interesting and novel, with therapeutic potential in pigs that are affected by a specific strain of *S. zooepidemicus*. Revision of the English in this manuscript is required, but generally the manuscript is well written and presented. Line 143: clarify challenge route and dose. This section states intravenous injection, but methods suggest that intraperitoneal challenge was used.

Line 151: statistical analysis should be performed. 12 of 15 tSzM survived compared with 2 of 10 controls ($p = 0.005$), 3 of 10 ST171 ($p = 0.03$) and 4 of 10 sezV ($p = 0.09$). Therefore, accounting for multiple measures, the tSzM was significantly more protected than the PBS, but not ST171 or sezV groups.

Line 154: present statistical analysis of bacterial burden results.

Line 258: should 'or' be 'and'? ie. program C involved three treatments with McAbs.

Line 197 and Figure 2D: could the anti-His tag McAb be interfering with binding of anti-cSzM-1 and anti-cSzM-3? Could anti-vSzM McAbs be dissociating from SzM? Treatment 4 results look incorrect based on the graph. This figure and the analysis should be revised.

Line 244, line 254 and line 278: present statistical analysis.

Line 278: Figure 5 shows that all PBS-treated mice died between 36 and 48 hours after infection with SEZ 17006, SEZ 18055 or SEE 17009. However, PBS-treated mice challenged with the 100-fold lower dose of SEZ ATCC35246 died between 72 and 96 hours after infection. Therefore, could the reduced effect of McAb treatment reflect the increased severity of the model using diverse strains of SEZ/SEE?

Lines 286-290 and 297 are not correct as a significant level of protection was only identified for anti-cSzM-1 McAb against SEZ 17006. Also remove 'wide' from lines 298 and 398.

Line 380-386: revise as cSzM-1 McAb provided partial protection against only SEZ 17006 challenge.

Discussion: Some discussion on the cost of McAb treatment of pigs should be provided. Is this really a practical approach to the treatment of pigs? SzM/SeM-based approaches have not proven to be efficacious in the natural host, the cost of McAb treatment of pigs is likely to be extremely high. Is administering McAb therapy within 4 hours of natural infection of pigs possible?

Line 420: please provide further details regarding humane endpoints and clinical monitoring of mice.

Figure 4: 'Progarm' should be 'Program'

Reviewer #2 (Comments for the Author):

Song et al. generated McAbs against different regions of the SzM protein of SEZ strain ATCC35246 and tested their protective efficacy in an intraperitoneal mouse model against homologous and heterologous challenges. These experiments included different time points of application of the McAb (prevention versus immunotherapy). The authors claim that application of McAb is promising for the treatment of streptococcosis in pigs (lines 33 to 38, lines 119 to 121, lines 400 to 402). Though the conclusions on the protective efficacy of the McAb in their animal model are justified, this perspective is very doubtful. There is not a single example for the usage of a murine McAb for the treatment of an infectious disease in porcine practice. The costs will be tremendous. Furthermore, it is not clear from the presented data if there is any therapeutic advantage in comparison to a classical antibiotic treatment which is much cheaper. Furthermore, the authors did not investigate functionality of murine McAbs in a porcine functional test such as an opsonophagocytosis test with porcine PMNs. Nevertheless, there is some merit in the comparative analysis of McAbs directed against conserved versus variable regions of SEZ including determination of affinity constants via SPR. This comparison is also of interest for a SzM-based active immunization. However, the novelty of the study is limited as SzM had already been shown to be a protective antigen (references 13 and 14) using a different SzM protein of an equine isolate and active rather than passive immunization. There are further critical points that need to be addressed by the authors which are specified below.

Major points:

1. Fig. 1A: Why only 7 days between prime and boost or boost and the second boost immunization?
2. Line 142 "lowest antibody titer increase rate": Immunogenicities of tSzM and the other vaccines were measured in ELISAs using different antigens (see M and M). The data cannot be compared, even more as CFA was used as adjuvant for priming in SzM immunization. How was the "antibody titer increase rate" calculated?
3. Lines 422 to 424: Mice were prime vaccinated with 200 μ g SzM in complete Freund's adjuvant. The use of Complete Freund's Adjuvant (CFA) in laboratory animals has the potential for causing severe pain and distress (see USDA Policy #11: Definition of Pain and Distress and Reporting Requirements for Laboratory Animals: Proceedings of the Workshop Held June 22, 2000). As rSzM is immunogenic and available in larger quantities after recombinant expression I doubt that the usage of CFA in this animal trial is in accordance with the published guidelines.
4. Fig. 2B: As the McAb are discussed to be used for the passive immunization of pigs, the authors should investigate or at least discuss the functionality of murine IgG1 McAb in a heterologous system, e.g. by measuring opsonophagocytosis after addition of IgG1 MAb against SzM to porcine blood infected in vitro with SEZ.
5. Lines 237 to 241: This conclusion ("McAb and PcAb - no complement activation function) is not justified based on the provided data. The authors should also analyze C3b deposition on the bacterial surface or use complement inhibitors or blood from C3 -/- mice to investigate the role of complement activation in more detail.
6. Lines 494 to 499: The complement hemolytic assay is described poorly though its results are crucial for the conclusion on the role of complement activation. Even if one looks at the provided reference the protocol remains nebulous. How were the McAb

used in this assay?

7. The cloning and expression of tSzM is not described appropriately (primer sequences are not provided, Fig. S1 is not sufficient)

8. The Results section contains subjective statements which need to be revised, example:

"In consideration of the extraordinarily high CFU used for the challenge, this protection rate was reasonable (lines 289 to 290)"

9. Statistical analysis is described poorly and only in the legends. Did the authors test the data set for normal distribution prior to application of the Student's test?

Staff Comments:

Preparing Revision Guidelines

Please return the manuscript within 60 days; if you cannot complete the modification within this time period, please contact me. If you do not wish to modify the manuscript and prefer to submit it to another journal, please notify me of your decision immediately so that the manuscript may be formally withdrawn from consideration by Microbiology Spectrum.

Response to reviewer comments:

Reviewer #1:

1. This manuscript describes the use of SzM and anti-SzM monoclonal antibodies against the variable and conserved domains of this protein to protect mice from infection with an *S. zooepidemicus* strain that was recovered from a case of infection in pigs. Variations of this protein have been shown previously to protect mice, but not horses from infection with *Streptococcus zooepidemicus* or *Streptococcus equi*. The application of monoclonal antibodies to treat infected animals is interesting and novel, with therapeutic potential in pigs that are affected by a specific strain of *S. zooepidemicus*. Revision of the English in this manuscript is required, but generally, the manuscript is well written and presented.

Response: I appreciate these positive estimates of our study. I have revised the English writing of this paper to make it easier to read and follow.

2. Line 143: clarify challenge route and dose. This section states intravenous injection, but methods suggest that intraperitoneal challenge was used.

Response: The intraperitoneal challenge was used after SzM protein immunization, whereas the intravenous injection was used in the evaluation of McAbs protective efficiency. I apologize that our interpretation leads to misunderstanding, we have revised the sentence of line 143 and the methods.

3. Line 151: statistical analysis should be performed. 12 of 15 tSzM survived compared with 2 of 10 controls ($p = 0.005$), 3 of 10 ST171 ($p = 0.03$) and 4 of 10 sezV ($p = 0.09$). Therefore, accounting for multiple measures, the tSzM was significantly more protected than the PBS, but not ST171 or sezV groups.

Response: We have added the statistical information of p values.

4. Line 154: present statistical analysis of bacterial burden results.

Response: Some of the CFU burdens in the same group are discrete and have a difference at the magnitude level, the statistical analysis shows no significance when comparing the mean values. So, in our interpretation, we emphasized the percentage of complete eradication of SEZ in organs instead of comparing mean values directly. The violin graph would be better to show the data distribution and avoid misleading the readers to compare the mean values.

5. Line 258: should 'or' be 'and'? ie. program C involved three treatments with McAbs.

Response: We have deleted the improper conjunction word “and”. Then use a semicolon instead of a comma to separate the sentences that describe three programs.

6. Line 197 and Figure 2D: could the anti-His tag McAb be interfering with the binding of anti-cSzM-1 and anti-cSzM-3? Could anti-vSzM McAbs be dissociating from SzM? Treatment 4 results look incorrect based on the graph. This figure and the analysis should be revised.

Response: The anti-His tag McAb was used as a positive control in this experiment. To address that the amount of anti-His tag McAb was not sufficient to saturate its epitope and will not interfere with the binding of anti-cSzM-1 and anti-cSzM-3 to this His-tag epitope (if they are anti-His rather than anti-SzM), we used this control antibody at both the beginning and the end of the interaction detection based on SPR. Depending on the additional data (treatment 6), the His tag McAb can still bind to its epitope and show a positive signal, indicating that the His-epitope was not saturated by treatment 3 or 4, the His-epitope was still available for anti-cSzM-1 and anti-cSzM-3 if they have an affinity to His-tag. Moreover, if the anti-His tag McAb added at the beginning (treatment 2) could interfere with the binding of the following antibodies added to the SPR system, it should also interfere with the binding of the anti-His tag McAb added at the endpoint (treatment 6). Together, we believe that the anti-His McAb did not interfere with the binding of anti-cSzM-1 and anti-cSzM-3.

There could be slight dissociation of antibodies in the interval wash step between each treatment. As shown in the anti-vSzM detection graph, though the chip was saturated by anti-vSzM-2 (treatment 1), after treatments 2 and 3, when added anti-vSzM-2 in treatment 4, the RU could still increase in a very short time and reach to saturation state very quickly. However, the RU signal increase rate of treatment 4 will not become as high as in treatments 2 and 3. So, we believe the slight dissociation will not leave huge gaps in the epitopes, and may not influence the binding detection of the following antibodies.

The RU change rate was calculated according to the change of RU value in the last 2/3 of the time range of each McAbs treatment, which could represent the stable stage of the interaction between two molecules. We have added the calculation details in the methods. Sorry for our neglect of this important information.

7. Line 244, line 254, and line 278: present statistical analysis.

Response: We have added the p values for statistical analysis of these data in the results part.

8. Line 278: Figure 5 shows that all PBS-treated mice died between 36 and 48 hours after infection with SEZ 17006, SEZ 18055, or SEE 17009. However, PBS-treated mice challenged with the 100-fold lower dose of SEZ ATCC35246 died between 72 and 96 hours after infection. Therefore, could the reduced effect of McAb treatment reflect the increased severity of the model using diverse strains of SEZ/SEE?

Response: In the cross-protection detection experiment, the 100% death time of the ATCC35246 challenged mice model (PBS group) is ranging from 72 hpi to 96 hpi, whereas the 100% death times of SEZ 17006, SEZ 18055 or SEE 17009 challenged mice models (PBS group) are ranging from ~36 hpi to 48 hpi. There is a ~24 hours difference, and we have to admit that we only consider the death percentage but neglect the 100% death time range, which may also influence the severity of the animal model. We have tried to make these animal models consistent, but the slight difference was very hard to eliminate, and had to sacrifice too many mice. So, we evaluated the cross-protection with animal models with a slight difference in severity. Nevertheless, these slight differences in animal models will not deny the protection of the McAbs against SEZ ATCC35246 infection in mice.

Below, we showed the additional database on the 10-fold lower dose SEE17009 challenged mice model, which had less severity (Fig 5E). However, both anti-cSzM and anti-vSzM McAbs still could not provide efficient protection, and there were abundant CFU in the organs of the only surviving mouse.

9. Lines 286-290 and 297 are not correct as a significant level of protection was only identified for anti-cSzM-1 McAb against SEZ 17006. Also, remove 'wide' from lines 298 and 398.

Line 380-386: revise as cSzM-1 McAb provided partial protection against only SEZ 17006 challenge.

Response: We have added more data in the detection of cross-protection of McAbs. With a lower challenge dose. According to these new data, we have rewritten the paragraphs about cross-protection evaluation in the abstract, results, and discussion parts to make the conclusion more rigorous.

10. Discussion: Some discussion on the cost of McAb treatment of pigs should be provided. Is this a practical approach to the treatment of pigs? SzM/SeM-based approaches have not proven to be efficacious in the natural host, the cost of McAb treatment of pigs is likely to be extremely high. Is administering McAb therapy within 4 hours of natural infection of pigs possible?

Response: We have added more interpretation to discuss the cost of McAbs treatment in animals, as well as the applicability of the McAbs administration programs in pigs or other animals. In addition, we agree that the murine isotype of McAbs produced in this research may not be administrated to pigs directly, so we tone down some of the conclusions and make them more rigorous. This research provided information for the development of chimeric McAbs or other genetically engineered McAbs that have potential application in protecting pigs against hypervirulent SEZ infection in the future.

11. Line 420: please provide further details regarding humane endpoints and clinical monitoring of mice.

Response: We have added more information.

12. Figure 4: 'Progarm' should be 'Program'

Response: Sorry for the spelling error. We have corrected the mistake in Figure 4.

Reviewer #2:

1. Song et al. generated McAbs against different regions of the SzM protein of SEZ strain ATCC35246 and tested their protective efficacy in an intraperitoneal mouse model against homologous and heterologous challenges. These experiments included different time points of application of the McAb (prevention versus immunotherapy). The authors claim that application of McAb is promising for the treatment of streptococcosis in pigs (lines 33 to 38, lines 119 to 121, lines 400 to 402). Though the conclusions on the protective efficacy of the McAb in their animal model are justified, this perspective is very doubtful. There is not a single example for the usage of a murine McAb for the treatment of an infectious disease in porcine practice. The costs will be tremendous. Furthermore, it is not clear from the presented data if there is any therapeutic advantage in comparison to a classical antibiotic treatment which is much cheaper. Furthermore, the authors did not investigate functionality of murine McAbs in a porcine functional test such as an opsonophagocytosis test with porcine PMNs.

Response: Thank you very much for your constructive advice and helpful comments. We have realized that some of the data of our research were overestimated, especially concerning the conclusion of further applications of the McAbs developed in this research. We agree that there is still a very long trip before the clinical application of these McAbs in pigs. So, we revised the conclusions to make them more rigorous. The mice McAbs isotype will be recognized as foreign antigen in pigs, and as we have toned down the conclusion about the direct porcine application of these McAbs, it may not be necessary for the further test with porcine PMNs. We also added more information and perspective about the high cost of using McAbs to treat infectious diseases, as well as the comparison of the McAbs and the classical antibiotic treatment in the discussion.

2. Nevertheless, there is some merit in the comparative analysis of McAbs directed against conserved versus variable regions of SEZ including determination of affinity constants via SPR. This comparison is also of interest for a SzM-based active immunization. However, the novelty of the study is limited as SzM had already been shown to be a protective antigen (references 13 and 14) using a different SzM protein of an equine isolate and active rather than passive immunization. There are further critical points that need to be addressed by the authors which are specified below.

Response: We appreciate the positive comments from the reviewer about the outcomes of the SRP analysis. We understand the reviewer's concern about the novelty of this study. The immune protection of the SzM protein from the horses-sourced SEZ strains has been investigated while SEZ was primarily considered a horse pathogen. Recently, the outbreak of SEZ emerged in the pig feeding industry of North America. Though the M family proteins in the group C streptococci have no systematic categorization method, the diversity of the SzM proteins between pigs and horses sourced strains have been identified. Our study focused on the SzM type from the pigs-sourced SEZ strain, which may have specific characteristics due to the diverse variable region sequence compared to the horse sourced strain. Besides, we considered the variable and conserved regions of this SzM separately for McAbs production and protective evaluation for the first time. Together, we believe that even though this research is not highly novel, it still has relative novelty and would contribute to the further investigation of preventive production development against pig sourced hypervirulent SEZ strain infection.

Major points:

3. Fig. 1A: Why only 7 days between prime and boost or boost and the second boost immunization?

Response: The P/N value ($P = \text{OD value of immunized serum}$, $N = \text{OD value of negative control serum}$) was the ratio of antibody titers detected with ELISA, when $P/N \geq 2.1$, the serum should be considered positive. With this 7-day interval immunization protocol, we found it was enough for all SzM protein immunized mice antibody titers that showed positive P/N values. So, we decide to use this immunization procedure.

4. Line 142 "lowest antibody titer increase rate": Immunogenicities of tSzM and the other vaccines were measured in ELISAs using different antigens (see M and M). The data cannot be compared, even more as CFA was used as adjuvant for priming in SzM immunization. How was the "antibody titer increase rate" calculated?

Response: We agree that these data cannot be compared because of the different coated antigens used in ELISA, and the term "antibody titer increase rate" was not rigorous. We have revised the interpretation.

5. Lines 422 to 424: Mice were prime vaccinated with 200 μg SzM in complete Freund's adjuvant. The use of Complete Freund's Adjuvant (CFA) in laboratory animals has

the potential for causing severe pain and distress (see USDA Policy #11: Definition of Pain and Distress and Reporting Requirements for Laboratory Animals: Proceedings of the Workshop Held June 22, 2000). As rSzM is immunogenic and available in larger quantities after recombinant expression I doubt that the usage of CFA in this animal trial is in accordance with the published guidelines.

Response: Thank you for pointing out this problem. We will pay more attention to the animal welfare issues in our animal experiments. According to the DSDA Policy #11, the Complete Freund's Adjuvant (CFA) used for antibody production may cause results ranging from momentary or slight pain to severe pain depending on the product, procedure, and species. Due to the high concentration of the SzM protein, we could minimize the volume of CFA used for the first immunization. Moreover, the injection process was conducted gently, the animals did not show stressful reactions during and after the injection. In the boost immunizations, we used incomplete Freund's adjuvant instead of the CFA.

6. Fig. 2B: As the McAb are discussed to be used for the passive immunization of pigs, the authors should investigate or at least discuss the functionality of murine IgG1 McAb in a heterologous system, e.g. by measuring opsonophagocytosis after addition of IgG1 MAb against SzM to porcine blood infected in vitro with SEZ.

Response: We realize that some of our perspectives about the application in pigs of the McAbs developed in this study are inappropriate. For heterologous application, further appropriate genetical modification based on these McAbs is necessary. We have revised the narrative to tone down our conclusion. So, hope the reviewer would allow that it is not necessary to add more data based on porcine blood.

7. Lines 237 to 241: This conclusion ("McAb and PcAb - no complement activation function) is not justified based on the provided data. The authors should also analyze C3b deposition on the bacterial surface or use complement inhibitors or blood from C3 -/- mice to investigate the role of complement activation in more detail.

Response: We have added more experiments to analyze the C3b deposition on the bacterial surface in Fig 3. The C3b deposition was not influenced by the addition of McAbs. We also make some revision on the description of the anti-SEZ biological function of McAbs.

8. Lines 494 to 499: The complement hemolytic assay is described poorly though its results are crucial for the conclusion on the role of complement activation. Even if

one looks at the provided reference the protocol remains nebulous. How were the McAb used in this assay?

Response: The complement hemolytic assay is designed as a control experiment to ensure that the complement in the serum can be activated and lyse sheep erythrocytes by inducing of the anti-sheep erythrocyte antibody. The McAbs produced in this research were not used in this assay, yet the antibody-mediated complement sterilization test only.

9. The cloning and expression of tSzM is not described appropriately (primer sequences are not provided, Fig. S1 is not sufficient)

Response: The primers' sequences information was added in Table S1 and their locations were marked in Figure S1A as well.

10. The Results section contains subjective statements which need to be revised, example: "In consideration of the extraordinarily high CFU used for the challenge, this protection rate was reasonable (lines 289 to 290)"

Response: We have added more data to this results part and revised the interpretation. The subjective statements had been deleted.

11. Statistical analysis is described poorly and only in the legends. Did the authors test the data set for normal distribution prior to application of the Student's test?

Response: We have added the statistical analysis p values in the manuscript. The details of statistical analysis were added to the M&M part as well. All data used for Student's t-test was tested for normal distribution.

September 15, 2022

Dr. Zhe Ma
Nanjing Agricultural University
Nanjing
China

Re: Spectrum01742-22R1 (Protection efficacy of monoclonal antibodies targeting different regions of specific SzM protein from swine-isolated *Streptococcus equi* ssp. *zooepidemicus* strains)

Dear Dr. Zhe Ma:

Your manuscript has been accepted, and I am forwarding it to the ASM Journals Department for publication. You will be notified when your proofs are ready to be viewed.

Sincerely,

Artem Rogovskyy
Editor, Microbiology Spectrum
